# Parental Prepuberty Overweight and Offspring Lung Function

**DOI:** 10.3390/nu14071506

**Published:** 2022-04-04

**Authors:** Marianne Lønnebotn, Lucia Calciano, Ane Johannessen, Deborah L. Jarvis, Michael J. Abramson, Bryndís Benediktsdóttir, Lennart Bråbäck, Karl A. Franklin, Raúl Godoy, Mathias Holm, Christer Janson, Nils O. Jõgi, Jorunn Kirkeleit, Andrei Malinovschi, Antonio Pereira-Vega, Vivi Schlünssen, Shyamali C. Dharmage, Simone Accordini, Francisco Gómez Real, Cecilie Svanes

**Affiliations:** 1Centre for International Health, Department of Global Public Health and Primary Care, University of Bergen, 5020 Bergen, Norway; ane.johannessen@uib.no (A.J.); jorunn.kirkeleit@uib.no (J.K.); cecilie.svanes@helse-bergen.no (C.S.); 2Department of Occupational Medicine, Haukeland University Hospital, 5021 Bergen, Norway; 3Unit of Epidemiology and Medical Statistics, Department of Diagnostics and Public Health, University of Verona, 37134 Verona, Italy; lucia.calciano@univr.it (L.C.); simone.accordini@univr.it (S.A.); 4Faculty of Medicine, National Heart & Lung Institute, Imperial College, London SW7 2AZ, UK; d.jarvis@imperial.ac.uk; 5MRC-PHE Centre for Environment and Health, Imperial College, London W2 1PG, UK; 6School of Public Health & Preventive Medicine, Monash University, Melbourne, VIC 3004, Australia; michael.abramson@monash.edu; 7Faculty of Medicine, University of Iceland, 101 Reykjavik, Iceland; brynben@hi.is; 8Section of Sustainable Health, Department of Public Health and Clinical Medicine, Umeå University, 901 87 Umeå, Sweden; lennart.braback@umu.se; 9The Department of Surgical and Perioperative Sciences, Surgery, Umeå University, 901 87 Umeaa, Sweden; karl.franklin@umu.se; 10Department of Pulmonary Medicine, University Hospital Complex of Albacete, University of Castilla La Mancha, 02008 Albacete, Spain; raul.godoymayoral@gmail.com; 11Occupational and Environmental Medicine, Institute of Medicine, School of Public Health and Community Medicine, Sahlgrenska Academy, University of Gothenburg, 405 30 Gothenburg, Sweden; mathias.holm@amm.gu.se; 12Department of Medical Sciences, Respiratory, Allergy and Sleep Research, Uppsala University, 751 85 Uppsala, Sweden; christer.janson@medsci.uu.se; 13Department of Clinical Science, University of Bergen, 5021 Bergen, Norway; nils.jogi@uib.no (N.O.J.); francisco.real@uib.no (F.G.R.); 14Department of Medical Sciences, Clinical Physiology, Uppsala University, 751 85 Uppsala, Sweden; andrei.malinovschi@medsci.uu.se; 15Pneumology Service, Juan Ramón Jiménez University Hospital in Huelva, 21005 Huelva, Spain; apv01h@gmail.com; 16Department of Public Health, Environment, Work and Health, Danish Ramazzini Centre, Aarhus University, 8000 Aarhus, Denmark; vs@ph.au.dk; 17National Research Center for the Working Environment, 2100 Copenhagen, Denmark; 18Allergy and Lung Health Unit, School of Population and Global Health, University of Melbourne, Melbourne, VIC 3010, Australia; s.dharmage@unimelb.edu.au; 19Department of Obstetrics and Gynecology, Haukeland University Hospital, 5053 Bergen, Norway

**Keywords:** lung function, overweight, nutrition, prepuberty, father/paternal/male/men, causal inference, intergenerational, counterfactual-based mediation analysis, ECRHS, RHINE, RHINESSA

## Abstract

In a recent study we found that fathers’ but not mothers’ onset of overweight in puberty was associated with asthma in adult offspring. The potential impact on offspring’s adult lung function, a key marker of general and respiratory health, has not been studied. We investigated the potential causal effects of parents’ overweight on adult offspring’s lung function within the paternal and maternal lines. We included 929 offspring (aged 18–54, 54% daughters) of 308 fathers and 388 mothers (aged 40–66). Counterfactual-based multi-group mediation analyses by offspring’s sex (potential moderator) were used, with offspring’s prepubertal overweight and/or adult height as potential mediators. Unknown confounding was addressed by simulation analyses. Fathers’ overweight before puberty had a negative indirect effect, mediated through sons’ height, on sons’ forced expiratory volume in one second (FEV_1_) (beta (95% CI): −144 (−272, −23) mL) and forced vital capacity (FVC) (beta (95% CI): −210 (−380, −34) mL), and a negative direct effect on sons’ FVC (beta (95% CI): −262 (−501, −9) mL); statistically significant effects on FEV_1_/FVC were not observed. Mothers’ overweight before puberty had neither direct nor indirect effects on offspring’s lung function. Fathers’ overweight starting before puberty appears to cause lower FEV_1_ and FVC in their future sons. The effects were partly mediated through sons’ adult height but not through sons’ prepubertal overweight.

## 1. Introduction

The paternal environment and exposures before conception on offspring health in humans have largely been neglected, despite animal models showing that the paternal environment plays a key role in non-genetic inheritance across generations through the male germline [1]. Paternal dietary factors, such as a high-fat diet [2,3,4] and a low-protein diet [5] have been found in murine models to be associated with offspring outcomes and with methylation patterns in sperm [6,7]. There are several potential windows of susceptibility during the lifespan of the father in which environmental effects could impact the epigenetic profile of his gametes, paternal prepuberty being one of them [8,9,10,11]. Human epidemiologic studies investigating such susceptibility windows are scarce, challenged by the temporal difficulties with covering two or more generations in humans. Furthermore, there is more data available on offspring’s health in relation to mothers’ than fathers’ environment and exposures.

A recent study from the Respiratory Health in Northern Europe, Spain and Australia (RHINESSA) cohort found that fathers’, but not mothers’, self-reported onset of overweight in puberty was associated with increased asthma risk in future adult offspring [12]. Three other studies from the same cohort found that fathers’, but not mothers’, smoking onset in puberty was associated with offspring’s asthma and reduced lung function [13,14,15]. Overweight and obesity are thought to be detrimental to lung function across age groups regardless of asthma status [16], and reduced lung function is a strong predictor of morbidity and mortality from non-communicable diseases, including chronic respiratory diseases in adult life [17]. The global increase in overweight [18], in parallel with the increase of asthma in children and adolescents, makes it important to further explore the effect of overweight on adverse lung health.

Studies suggest that childhood overweight and obesity influences growth patterns and pubertal development [19,20], and that height might be influenced by the timing of pubertal events [21]. Height is of particular importance when respiratory health is considered because it is related to lung growth and lung volumes [22]. In the present study, we used statistical models aimed at approaching a causal interpretation, to study the potential effects of parents’ overweight before puberty (generation G0) on adult offspring’s lung function (generation G1). The paternal and maternal lines were analysed separately, we examined whether effects were mediated through offspring’s own overweight before puberty or offspring’s adult height, and we analysed the potential moderating effect of offspring’s sex on these relationships to understand potential differences between sons and daughters.

## 2. Materials and Methods

### 2.1. Study Design

The European Community Respiratory Health Survey (ECRHS, www.ecrhs.org is an international, population-based cohort study of respiratory health in subjects aged 20–44 years at the time of recruitment (ECRHS I; 1992–1994) in 56 study centres from 25 countries (approximately 3000 subjects per study centre) [23,24]. In the seven Northern European centres (Aarhus in Denmark; Tartu in Estonia; Reykjavik in Iceland; Bergen in Norway; Gothenburg, Umeå and Uppsala in Sweden) all responders to the 1992 postal survey were followed in a large longitudinal questionnaire study, the Respiratory Health in Northern Europe (RHINE; www.rhine.nu) study [25]. Both the ECRHS and RHINE studies conducted follow-up studies in approximately 2002 (ECRHS II/RHINE II) and again in approximately 2012 (ECRHS III/RHINE III) [26].

The Respiratory Health in Northern Europe, Spain and Australia (RHINESSA, www.rhinessa.net) generation study examines the offspring of the participants in ten ECRHS/RHINE centres (in addition to the seven Northern European centres, two centres in Spain (Albacete and Huelva) and one in Australia (Melbourne)). Extensive questionnaires and lung function data were collected in clinical centres in the period 2013 to 2016. The RHINESSA study protocols were harmonised with the ECRHS protocols.

Informed consent was obtained from each participant, and the appropriate regional committees of medical research ethics approved each survey of the ECRHS, RHINE and RHINESSA studies.

### 2.2. Study Population

A flow chart describing the study population is presented in Figure 1. Of the 1405 adult offspring (aged 18 years or greater) who participated in the clinical stage of the RHINESSA study, 1025 had a parent who had participated in the most recent study.

ECRHS/RHINE follow-up studies (2010–2013). The database includes one parent only per offspring, resulting in two eligible study populations for the present analysis: one population for the paternal line (439 offspring and 322 fathers) and one population for the maternal line (586 offspring and 434 mothers). The number of offspring with valid lung function measurements and complete information on key variables was 420 for the paternal line and 510 for the maternal line, originating from 308 fathers and 388 mothers, respectively. These offspring and parents were included in our analysis. Distribution by RHINESSA study centre in the paternal and maternal lines is shown in online Appendix A.

### 2.3. Lung Function and Definitions

At RHINESSA clinical examinations, the maximum pre- and post-bronchodilator forced expiratory volume in one second (FEV_1_) and forced vital capacity (FVC) were recorded as the best of at least two manoeuvres that met the American Thoracic Society criteria for repeatability [27]. On the same occasion, the offspring’s height was objectively measured.

Overweight status was defined by using a validated figural drawing scale of nine sex-specific body silhouettes [28] (Figure 2). Parents in the ECRHS/RHINE III study were asked to tick the figural scale that best described their figure at specific time points including age eight years, voice break/menarche and age 30 years. Participants were classified as overweight if their self-reported body silhouettes were equal to or higher than figure four for females and figure five for males. In detail, ECRHS parents’ overweight status was classified as “overweight before puberty”, i.e., at age 8 years and/or in puberty (voice break for fathers and menarche for mothers), “overweight at age 30 years but not before puberty”, i.e., at age 30 years but neither at age 8 years nor in puberty, and “never overweight”, i.e., neither at age 8 years nor in puberty nor at age 30 years. RHINESSA offspring’s “overweight before puberty” (present vs. absent), i.e., at age eight years and/or in puberty, were reported by the adult offspring using the same figural drawing scale described above for their parents. Offspring reported their own smoking history (never or ever smoker).

ECRHS/RHINE parents’ education level was categorised as “low” if less than or equal to the minimum school-leaving age [29].

### 2.4. Statistical Analysis

Counterfactual-based mediation analyses [30] were carried out to investigate the pathways between parents’ overweight before puberty and offspring’s lung function. The mediation analysis allows decomposing the total effect of the exposure on the outcome into the natural indirect effect (i.e., acting through the mediators) and the natural direct effect (i.e., not acting through the mediators). The main requirement for mediation is that the indirect effect is statistically significant under the assumption of no unmeasured confounding. In our analysis, mediation was combined with moderation (moderated mediation) to determine whether the indirect effect varied across levels of the moderator variable [31].

Two multi-group mediation models [32] were used within the paternal and maternal lines. Model 1 included offspring’s FEV_1_ and FVC as the normally distributed, correlated, parallel outcomes. Model 2 included offspring’s FEV_1_/FVC as the normally distributed outcome. In both models, fathers’ and mothers’ overweight before puberty was the exposure of interest and a serial causal chain of the two mediators (offspring’s overweight before puberty and offspring’s height) was assumed, since we hypothesised a causal order between the mediators. In addition, parents’ education level was the potential confounding variable of the exposure–mediator, mediator–mediator and mediator–outcome relationships; offspring’s age and their own smoking history were analysed as adjustment variables of the exposure–outcome and mediator–outcome relationships. The moderator (offspring’s sex) was used to separate the observations into subgroups (sons and daughters). Figure 3 (model 1) and online Appendix A (model 2) provide a graphic depiction of all directional paths of the hypothesised mediation models.

Due to the non-normality of offspring’s overweight before puberty, the magnitude of the natural (counterfactual-based) direct and indirect effects was computed based on the latent response variable mediator approach with probit link, theta parameterisation and weighted least square mean and variance adjusted (WLSMV) estimators [33] with robust standard errors (father or mother = cluster variable). The WLSMV estimator yielded probit regression coefficients for the effects on the latent mediator (offspring’s overweight before puberty) and linear regression coefficients for the effects on offspring’s adult height, FEV_1_, FVC and FEV_1_/FVC. Centre clustering of our data was not considered (intraclass correlation < 0.1) due to the complexity of the relationships. The natural direct and indirect effects of the exposure on the outcomes were summarised as differences in offspring’s expected lung function values. The natural direct effect is the difference in offspring’s expected lung function value for the change in exposure status, keeping offspring’s height and/or offspring’s overweight at their expected value when the exposure is absent. The natural indirect effect is the difference in offspring’s expected lung function value when the exposure is present, but the offspring’s height and/or offspring’s overweight change from their expected value when the exposure is absent to their expected value when the exposure is present. Non-bias-corrected bootstrap 95% confidence intervals (95% CI) (10,000 resamples) were obtained for the causally defined effects in order to take their asymmetric distribution into account. In order to test the difference (Δ) of direct and indirect effects between sons and daughters, the non-bias-corrected bootstrap CI for the group difference in the direct and indirect effects [26] was computed.

#### Sensitivity Analyses

Using the Umediation R package (https://github.com/SharonLutz/Umediation), we assessed whether the estimated direct and indirect effects change after the inclusion of up to two unmeasured confounders for the exposure–outcome, exposure–mediator and mediator–outcome relationships. We simulated two normally distributed unmeasured confounders (U1 and U2) with a mean of 0 and variance of 0.001, within a single-exposure, single-mediator and single-outcome framework. As inputs for Umediation, we used the coefficients of the mediation analysis. The simulation analyses were carried out under multiple scenarios for the effects (beta regression coefficients) of the unmeasured confounder “U_1_” on the outcome (beta_U__→__o_), the mediator (beta_U__→__M_), and the exposure (beta_U__→__E_) by fixing beta_U__→__E_ = beta_U__→__M_ = beta_U__→__o_ = 0, 1, 3, 5, 7 and 9. For this purpose, pre-bronchodilator FEV_1_ and FVC were expressed in decilitres. We repeated the simulations by adding “U_2_” to the models under the same assumptions. We specified 1000 simulation runs and 1000 Monte Carlo draws for the nonparametric bootstrap.

Furthermore, we assessed how our results changed when offspring’s post-bronchodilator lung function measurements (available from 191 sons and 209 daughters in the paternal line, and from 222 sons and 255 daughters in the maternal line) were used as outcomes.

STATA 16 (StataCorp, College Station, TX, USA), Mplus 8.6 (Muthén and Muthén, Los Angeles, CA, USA) and R 3.6.1 (www.R-project.org) were used for the statistical analyses.

## 3. Results

### 3.1. Main Characteristics of the Study Subjects

The median age of fathers and mothers at participation in ECRHS/RHINE III was 56 and 55 years, respectively. The median age of adult offspring was 29 years (53.1% daughters) in the paternal line (Table 1a) and 31 years (54.5% daughters) in the maternal line (Table 1b). A total of 12% of the 308 fathers reported being overweight before puberty, compared to 25% of the 388 mothers. Furthermore, 10.4% of the fathers and 22.9% of the mothers reported being overweight at 30 years old, starting after puberty. Overweight before puberty was reported by 21.7% of the offspring in the paternal line and by 20.4% of the offspring in the maternal line, and more frequently by female offspring (*p*-value < 0.001). In both paternal and maternal lines, daughters had pre- and post-bronchodilator FEV_1_ and FVC values that were statistically significantly lower (*p*-value < 0.001), and pre- and post-bronchodilator FEV_1_/FVC that were statistically significantly higher (*p*-value < 0.001) than sons. Ever smoking was reported by 25.2% and 32.6% of the offspring in the paternal and maternal lines, respectively.

### 3.2. Mediation Analysis

#### 3.2.1. Paternal Line

Fathers’ overweight before puberty had a negative direct effect on sons’ adult height (beta (95% CI): −3.42 (−6.18, −0.57) cm), and sons’ adult height, in turn, had a positive direct effect on their own FEV_1_ (beta (95% CI): 42 (31, 53) mL) and FVC (beta (95% CI): 61 (48, 74) mL) (Table 2). Furthermore, fathers’ overweight before puberty had a positive direct effect on daughters’ overweight before puberty (beta (95% CI): 0.83 (0.32, 1.45)), which in turn had a negative direct effect on their own height in adulthood (beta (95% CI): −1.17 (−2.28, −0.09) cm) (Table 2).

Fathers’ overweight before puberty had both a negative direct effect (Table 2) and a negative indirect effect through offspring’s height on offspring’s FVC (Table 3) (complementary mediation), compared to fathers who had never been overweight. The direct effect on offspring’s FVC was moderated by offspring’s sex (Δ (95% CI): −340 (−642, −36) mL); it was statistically significant in sons (beta (95% CI): −262 (−501, −9) mL) but not in daughters. The indirect effect was statistically significant only in sons (beta (95% CI): −210 (−380, −34) mL) and no moderated mediation by offspring’s sex was observed (Δ (95% CI): −123 (−319, 69) mL).

In addition, fathers’ overweight before puberty had statistically significant indirect (Table 3) but not direct (Table 2) effects on offspring’s FEV_1_ (indirect-only mediation), compared to fathers who had never been overweight. Specifically, this indirect effect occurred in sons through their own adult height (beta (95% CI): −144 (−272, −23) mL) but not in daughters (Table 3). However, no moderated mediation by offspring’s sex was observed because the difference in indirect effects between sons and daughters (Δ (95% CI): −81 (−230, 54) mL) was not statistically significant. We did not observe statistically significant effects of fathers’ overweight before puberty on offspring’s FEV_1_/FVC (online Appendix A).

#### 3.2.2. Maternal Line

Mothers’ overweight before puberty had neither direct nor indirect effects (mediated by offspring’s overweight before puberty and/or offspring’ adult height) on offspring’s lung function (non-mediation; Table 4 and Table 5; online Appendix A).

### 3.3. Sensitivity Analyses

The potential role of unmeasured confounders was assessed by Umediation analysis. This was only conducted for sons’ FEV_1_ and FVC in the paternal line. The inclusion of two unmeasured confounders (online Appendix A) in the model had a limited impact on the direct and indirect effects of fathers’ overweight before puberty on adult sons’ FEV_1_ and FVC, also when each unmeasured confounder had a very strong effect on the outcome, mediator, and exposure. In fact, when beta_U__→__E_, beta_U__→__M_ and beta_U__→__O_ were set less than or equal to five, the proportion of simulations where the results matched (whether U_1_ and U_2_ were included or excluded from the models) was greater than 80.1% and the average absolute difference of the average effects was lower than 0.176.

When offspring’s post-bronchodilator values were used, the indirect effect of fathers’ overweight before puberty on FEV_1_ (beta (95% CI): −165 (−312, −29) mL) and FVC (beta (95% CI): −223 (−410, −41) mL) in sons remained statistically significant (Appendix A). We found nonmediation in the maternal line (Appendix A).

## 4. Discussion

Using statistical models for causal inference, we found that fathers’ overweight may negatively affect adult sons’ lung function. Our results suggest that the sons of fathers who were overweight before puberty had substantially lower FEV_1_ and FVC compared to the sons of fathers who were never overweight. The causal association of fathers’ overweight before puberty with FEV_1_ and FVC was, respectively, fully (indirect-only effect) and partially (involving both direct and indirect effects) mediated through sons’ adult height, but not through sons’ own overweight before puberty. In the maternal line, no direct and indirect effects between mothers’ overweight status and adult offspring’s lung function were found. Using mediation analyses, we were able to show that the observed association between a father’s overweight before puberty and adult offspring’s lung function involved mediation through lower adult-attained height in his sons. Our results support the concept that the metabolic environment in male prepuberty might influence the health of the next generation.

To our knowledge, this is the first study investigating parental overweight long before conception and adult offspring’s lung function. In a study by Accordini et al. [15], causal effects of tobacco smoking on offspring lung function in the paternal line were investigated and they found that fathers’ smoking initiation in prepuberty may cause lower lung function in offspring. This is the only other human study we know of that investigates exposure in paternal prepuberty in relation to offspring lung function. There are inter-and transgenerational studies investigating exposures in mid-childhood and sex-specific effects. Johannessen, Calciano, Lønnebotn et al. [12] found that the onset of overweight in paternal puberty was an important risk factor for their adult offspring’s asthma without nasal allergies, while no such associations were found in the maternal line. Analyses of the ALSPAC cohort found that fathers’ onset of smoking at <11 years was related to increased fat mass in their adolescent sons [34]. The historical Överkalix study showed that paternal grandfathers’ food availability during prepuberty was associated with the mortality rate of the grandsons [35].

Multiple animal studies indicate that dietary conditions can induce epigenetic changes through the male germline. These changes can be transferred to the embryo, inducing phenotypic or metabolic perturbations in the offspring [36]. A parental high-fat diet in rats has been shown to affect offspring even if the exposure takes place before in utero development [2] and a paternal high-fat diet has been associated with impaired glucose tolerance and increased body weight in rat offspring [3]. A rodent study demonstrated that male offspring from obese fathers fed a high-fat diet for 12 weeks before mating with control fed females had reduced birth weight and a growth deficit phenotype was observed from birth to 6 months of age [4]. Both a high-fat diet and a low-protein diet have been linked to altered methylation patterns in sperm cells [5,6,7]. These data indicate that dietary conditions during male gametogenesis can influence the metabolic status of the offspring and affect phenotypic outcomes [8].

We questioned whether offspring’s overweight in prepuberty could affect their own height in adulthood. Based on existing literature, adult height might be influenced by the timing of pubertal events, and the onset of puberty could be related to overweight and obesity [19,21,22,37]. We found in our analysis a positive direct effect of fathers’ overweight in prepuberty on daughters’ overweight before puberty, which in turn had a negative direct effect on their own height in adulthood. Despite this association, fathers’ overweight in prepuberty was neither directly nor indirectly associated with daughters’ lung function in adulthood.

Fathers’ overweight before puberty reduced both sons’ and daughters’ height in adulthood, compared to fathers who had never been overweight, although it was only statistically significant in sons. Furthermore, the mediated effect through lower adult-attained height on offspring’s FEV_1_ and FVC was only found in sons. In the maternal line, mothers’ overweight status in different time periods was not associated with offspring’s height in adulthood. Height is genetically inherited, but independent of genetics and a shared postnatal environment, paternal influences acting through epigenetic mechanisms could possibly affect offspring’s growth and height [38].

Height is of particular importance when respiratory health is considered because it is related to lung growth and lung volume [22]. Our findings raise a broader issue around the common practice of adjusting for height in analyses using absolute lung function values. In our analyses, height partly mediated the relationship between fathers’ prepuberty overweight and lung function outcomes in their sons. This gives important additional understanding and would not have been possible to disentangle if we had adjusted for height or used transformations of lung function outcomes (e.g., z-score or % predicted), methods that inherently take differences in the attained height into account [39].

The effects of fathers’ overweight on son’s lung function were most consistent for FVC. FVC is an important predictor of all-cause mortality in asymptomatic adults without chronic respiratory conditions [17]. Furthermore, it has been shown that early life factors and genetic effects that manifest in childhood may influence the individual’s lifespan FVC and FEV_1_ trajectories [40,41]. There is substantial epidemiological and experimental evidence supporting the concept of developmental origins [42] of adult lung disease and low lung function [43]. There is strong evidence of an association between birth weight and adult FVC [44] and we speculate that birth weight partly could be a mediator for paternal factors occurring before conception. The investigation of early life factors has traditionally been focused on how maternal exposures might influence offspring’s health and risk of disease, with the in utero environment as one particular susceptibility window [42]. Nevertheless, paternal influences acting on the gamete environment long before the conception of the child could possibly affect foetal conditions and development in utero, a function of the placenta and foetal growth [9,45,46].

A major strength of this study is the statistical approach used for assessing causal associations to estimate causal effects across two generations. Counterfactual mediation models enabled us to simultaneously investigate all the mediation paths (via multiple ordered mediators) in the same model. Further, we used probabilistic simulations to estimate the impact of unmeasured confounding, which suggested that there were not important effects of unmeasured confounding such as genetic or unknown environmental factors in any of the two generations.

The data available from the RHINESSA study represents another important strength. Offspring’s lung function, weight and height were objectively measured at a clinical visit. The RHINESSA study contains comprehensive and comparable information on multiple generations, and both the offspring and the parent generations have been investigated following similar protocols. Furthermore, the RHINESSA study offers detailed information not only on mothers, but also on fathers, and there are data on parental puberty which is rarely available in multigenerational human cohorts.

We used figural drawing scales to assess body size. These identified subjects as “at risk” for overweight body size. A validation study identified the optimal cut-offs that we used to identify overweight in adulthood (25 ≤ BMI < 30 kg/m^2^) [28]. The figural drawing scale tool has been validated against measured and self-reported height and weight, both with regard to the current [28] and past body silhouettes [47]; however, we have not been able to validate the body silhouettes from childhood and puberty ages. The same figural drawing scales were used for different ages and thereby allowed for direct comparison across these age periods, enabling us to identify the important window of susceptibility in parents for respiratory health in offspring. Further, the same figural drawing scales were available for both parents and offspring; thus, the analytical models could include comparable body size measures in parents (exposure) and offspring (mediator).

## 5. Conclusions

Fathers’ overweight starting before puberty appears to cause lower lung function in adult sons. The effects seem to be partially mediated by the offspring’s height, but not by the offspring’s own overweight. We did not find causal associations in the maternal line. Our findings support the concept that the metabolic environment in males’ prepuberty might influence the health of the next generation. Increased scientific attention to male puberty in relation to future generations’ health may have profound implications and open new opportunities for targeted public health strategies. We speculate that one might improve the health of two generations while intervening in one generation—in an age window that is key for both.

## Figures and Tables

**Figure 1 nutrients-14-01506-f001:**
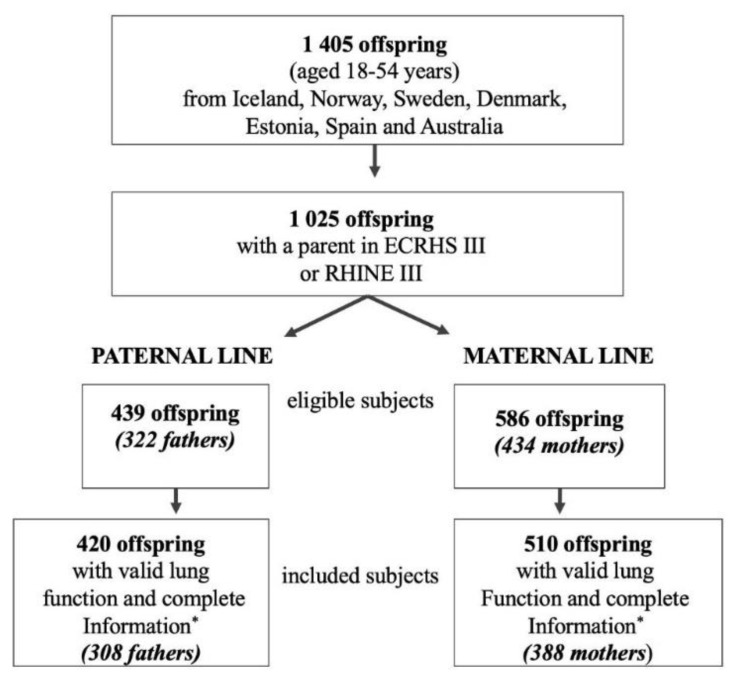
Study population RHINESSA clinical-stage flow chart. * Offspring with information on their own body silhouette and height and on body silhouettes of their participating parent.

**Figure 2 nutrients-14-01506-f002:**
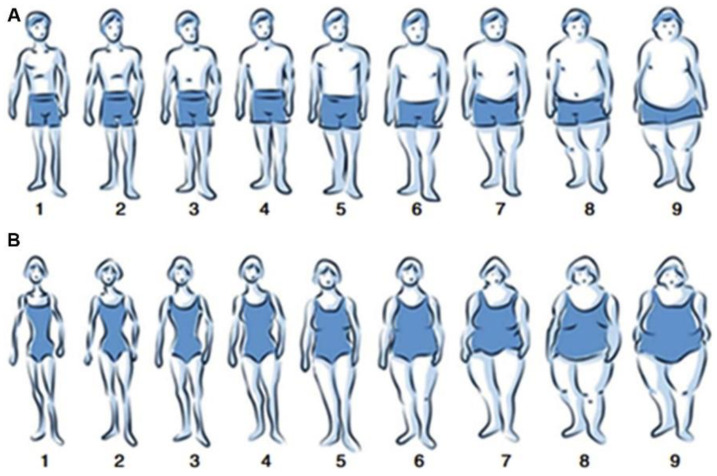
Body silhouettes for men (**A**) and women (**B**) used in the ECRHS/RHINE III studies and in the RHINESSA questionnaire survey. Cut-offs for overweight status were five or greater in men, and four or greater in women.

**Figure 3 nutrients-14-01506-f003:**
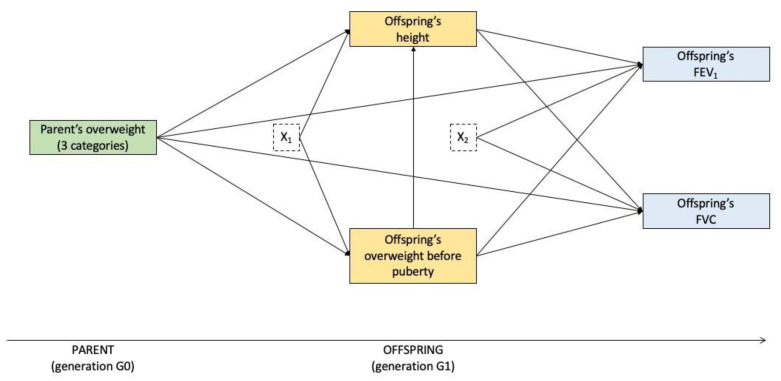
Graphical representation of the mediation model for FEV_1_ and FVC in sons or daughters within the paternal or maternal lines (model one). The green box represents the exposure of interest, the yellow boxes the mediators and the blue boxes the outcomes. The dotted boxes represent the set of potential confounders and adjusting variables of the mediators (X_1_: parents’ education level) and of the outcomes (X_2_: parents’ education level and offspring’s age and their own smoking history).

**Table 1 nutrients-14-01506-t001:** (**a**) Main characteristics of fathers and offspring in the paternal line. (**b**) Main characteristics of mothers and offspring in the maternal line.

**(a)**					
		**Paternal Line**			
**Generation**		**Total**	**Sons**	**Daughter’s**	***p*-Value** **^‡^**
Father (G0)	Number of fathers	308	165	143	-
	Age (years), median (range)	56 (40–66)	55 (40–66)	56 (40–65)	0.311
	Low education level ^†^, % (n)	8.4 (26)	8.5 (14)	8.4 (12)	0.571
	Overweight status, % (n)				
	before puberty	12.0 (37)	12.1 (20)	11.9 (17)	0.747
	at age 30 years but not before puberty	10.4 (32)	9.1 (15)	11.9 (17)	
	never	77.6 (239)	78.8 (130)	76.2 (109)	
Offspring (G1)	Number of adult offspring	420	197	223	-
	Age (years), median (range)	29 (18–51)	30 (18–47)	28 (18–51)	0.122
	Height (cm), mean (sd)	174.1 (8.98)	181 (6.5)	168 (6.3)	<0.001
	Pre-bronchodilator FEV_1_ (mL), mean (sd)	3798 (761)	4387 (629)	3277 (405)	<0.001
	Pre-bronchodilator FVC (mL), mean (sd)	4692 (969)	5490 (727)	3987 (492)	<0.001
	Pre-bronchodilator FEV_1_/FVC%, mean (sd)	81.3 (6.0)	80.0 (6.0)	82.4 (5.8)	<0.001
	Post-bronchodilator FEV_1_ * (mL), mean (sd)	3918 (790)	4531 (648)	3359 (394)	<0.001
	Post -bronchodilator FVC * (mL), mean (sd)	4673 (978)	5460 (750)	3954 (481)	<0.001
	Post-bronchodilator FEV_1_/FVC% *, mean (sd)	84.2 (5.7)	83.1 (5.6)	85.2 (5.6)	<0.001
	Overweight before puberty, % (n)	21.7 (91)	12.7 (25)	29.6 (66)	<0.001
	Ever smoking, % (n)	25.2 (106)	28.4 (56)	22.4 (50)	0.177
**(b)**					
		**Maternal Line**			
**Generation**		**Total**	**Sons**	**Daughter’s**	***p*-Value** **^‡^**
Mother (G0)	Number of mothers	388	197	191	-
	Age (years), median (range)	55 (40–66)	55 (40–66)	55 (40–66)	0.869
	Low education level ^†^, % (n)	10.1 (39)	8.6 (17)	11.5 (22)	0.219
	Overweight status, % (n)				
	before puberty	25.0 (97)	24.9 (49)	25.1 (48)	0.910
	at age 30 years but not before puberty	22.9 (89)	23.9 (47)	22.0 (42)	
	never	52.1 (202)	51.2 (101)	52.9 (101)	
Offspring (G1)	Number of adult offspring	510	232	278	-
	Age (years), median (range)	31 (18–54)	31 (18–46)	30 (18–54)	0.797
	Height (cm), mean (sd)	173.9 (9.44)	181 (6.4)	168 (6.6)	<0.001
	Pre-bronchodilator FEV_1_ (mL), mean (sd)	3780 (814)	4430 (620)	3238 (500)	<0.001
	Pre-bronchodilator FVC (mL), mean (sd)	4675 (1028)	5537 (761)	3957 (568)	<0.001
	Pre-bronchodilator FEV_1_/FVC%, mean (sd)	81.1 (5.9)	80.2 (5.9)	81.9 (5.9)	<0.001
	Post-bronchodilator FEV_1_ * (mL), mean (sd)	3915 (832)	4577 (630)	3339 (488)	<0.001
	Post -bronchodilator FVC * (mL), mean (sd)	4679 (1039)	5526 (785)	3942 (561)	<0.001
	Post-bronchodilator FEV_1_/FVC% *, mean (sd)	84.0 (5.5)	83.1 (5.6)	84.8 (5.4)	<0.001
	Overweight before puberty, % (n)	20.4 (104)	17.7 (41)	22.7 (63)	0.186
	Ever smoking, % (n)	32.6 (166)	30.2 (70)	34.5 (96)	0.299

^†^ Less than or equal to the minimum school-leaving age. (a) * Available from 191 sons and 209 daughters in the paternal line. (b) * Available from 222 sons and 255 daughters in the maternal line. ^‡^ *p*-values obtained by using a two-sample *t*-test, two-sample test on the equality of medians, χ^2^ test or Fisher exact test.

**Table 2 nutrients-14-01506-t002:** Natural direct effects * on offspring’s pre-bronchodilator FEV_1_ and FVC (model 1) within the paternal line.

	Offspring’s Overweight before Puberty ^‡^ Beta (95% CI)		Offspring’s Adult Height (cm) Beta (95% CI)		Offspring’s FEV_1_ (mL) Beta (95% CI)		Offspring’s FVC (mL) Beta (95% CI)	
	Sons	Daughters	Sons	Daughters	Sons	Daughters	Sons	Daughters
Fathers’ overweight (vs. never)								
before puberty	0.56 (−0.19, 1.19)	**0.83** **(0.32, 1.45)**	**−3.42** **(−6.18, −0.57)**	−2.11 (−4.58, 0.65)	−164 (−355, 45)	26 (−155, 187)	**−262** ^†^ **(−501, −9)**	78 ^†^ (−138, 283)
at age 30 years but not before								
puberty	0.07 (−4.26, 0.88)	−0.02 (−0.79, 0.57)	−0.17 (−3.56, 6.85)	−0.2 (−2.72, 1.87)	−15 (−313, 504)	51 (−101, 209)	−43 (−363, 632)	64 (−109, 252)
Offspring’s overweight before puberty (vs. absent)	-	-	0.37 (−1.16, 1.99)	**−1.17** **(−2.28, −0.09)**	36 (−64, 139)	32 (−35, 97)	54 (−51, 170)	48 (−29, 122)
Offspring’s height in adulthood (cm)	-	-	-	-	**42** **(31, 53)**	**30** **(21, 40)**	**61 ^†^** **(48, 74)**	**41 ^†^** **(31, 53)**

* Difference (beta) in offspring’s expected lung function value for the change in exposure status, keeping offspring’s height and/or offspring’s overweight at their expected value when the exposure is absent. Model 1 also includes the potential confounders and adjusting variables of the mediators (fathers’ low education level) and of the outcomes (fathers’ low education level and offspring’s age and ever smoking). Beta is a probit regression coefficient for the effect on the latent mediator (offspring’s overweight before puberty) and a linear regression coefficient for the effect on offspring’s adult height, FEV_1_ and FVC. ^‡^ Offspring’s overweight before puberty was considered as a continuous latent mediator variable. ^†^ The difference in direct effects between sons and daughters is statistically significant at *p* < 0.05. 95% CI: 95% confidence interval. The statistically significant effects are indicated in **bold.**

**Table 3 nutrients-14-01506-t003:** Natural indirect effects * on offspring’s pre-bronchodilator FEV_1_ and FVC (model 1) within the paternal line.

		Offspring’s FEV_1_ (mL) Beta (95% CI)		Offspring’s FVC (mL) Beta (95% CI)	
	Indirect Effects	Sons	Daughters	Sons	Daughters
Father’s overweight (vs. never)					
before puberty	via offspring’s overweight	20 (−41, 111)	26 (−30, 99)	30 (−30, 143)	40 (−24, 126)
	via offspring’s height	**−144** **(−272, −23)**	−64 (−146, 19)	**−210** **(−380, −34)**	−87 (−202, 26)
	via offspring’s overweight and height	9 (−31, 69)	−29 (−83, 1)	13 (−43, 99)	−40 (−111, 2)
at age 30 years but not before puberty	via offspring’s overweight	3 (−426, 69)	−1 (−35, 33)	4 (−555, 79)	−1 (−47, 43)
	via offspring’s height	−7 (−152, 267)	−6 (−86, 58)	−10 (−218, 395)	−8 (−118, 77)
	via offspring’s overweight and height	1 (−229, 47)	1 (−26, 34)	2 (−336, 66)	1 (−36, 46)

* Difference (beta) in offspring’s expected lung function value when the exposure is present, but offspring’s height and/or offspring’s overweight change from their expected value when the exposure is absent to their expected value when the exposure is present. Model 1 also includes the potential confounders and adjusting variables of the mediators (fathers’ low education level) and of the outcomes (fathers’ low education level and offspring’s age and ever smoking). 95% CI: 95% confidence interval. The statistically significant effects are indicated in **bold.**

**Table 4 nutrients-14-01506-t004:** Natural direct effects * on offspring’s pre-bronchodilator FEV_1_ and FVC (model 1) within the maternal line.

	Offspring’s Overweight before Puberty ^‡^, Beta (95% CI)		Offspring’s Adult Height (cm) Beta (95% CI)		Offspring’s FEV_1_ (mL) Beta (95% CI)		Offspring’s FVC (mL) Beta (95%CI)	
	Sons	Daughters	Sons	Daughters	Sons	Daughters	Sons	Daughters
Mothers’ overweight (vs. never)								
before puberty	0.59 (−0.31, 0.82)	0.52 (−0.09, 0.71)	0.1 (−3.54, 1.13)	−0.12 (−3.25, 0.82)	115 (−182, 199)	160 (−39, 213)	124 (−237, 229)	110 (−117, 170)
at age 30 years but not before puberty	0.54 (−0.34, 0.78)	0.3 (−0.45, 0.5)	1.41 (−2.33, 2.5)	1.1 (−2.4, 2.04)	98 (−228, 200)	186 (−47, 244)	116 (−263, 236)	180 (−87, 253)
Offspring’s overweight before puberty (vs. absent)	-	-	1.38 (−0.77, 1.96)	0.57 (−1.31, 1.09)	114 (−69, 165)	82 (−36, 113)	35 (−69, 192)	83 (−51, 117)
Offspring’s height in adulthood (cm)	-	-	-	-	**53** **(34, 59)**	**42** **(29, 46)**	**75** **^†^** **(53, 82)**	**53** **^†^** **(38, 58)**

* Difference (beta) in offspring’s expected lung function value for the change in exposure status, keeping offspring’s height and/or offspring’s overweight at their expected value when the exposure is absent. Model 1 also includes the potential confounders and adjusting variables of the mediators (mothers’ low education level) and of the outcomes (mothers’ low education level and offspring’s age and ever smoking). Beta is a probit regression coefficient for the effect on the latent mediator (offspring’s overweight before puberty) and a linear regression coefficient for the effect on offspring’s adult height, FEV_1_ and FVC. ^‡^ Offspring’s overweight before puberty was considered as a continuous latent mediator variable. ^†^ The difference in direct effects between sons and daughters is statistically significant at *p* < 0.05. 95% CI: 95% confidence interval. The statistically significant effects are indicated in **bold.**

**Table 5 nutrients-14-01506-t005:** Natural indirect effects * on offspring’s pre-bronchodilator FEV_1_ and FVC (model 1) within the maternal line.

		Offspring’s FEV_1_ (mL) Beta (95% CI)		Offspring’s FVC (mL) Beta (95% CI)	
	Indirect Effects	Sons	Daughters	Sons	Daughters
Mother’s overweight (vs. never)					
before puberty	via offspring’s overweight	34 (−24, 70)	27 (−10, 49)	40 (−29, 81)	28 (−14, 51)
	via offspring’s height	4 (−160, 48)	−4 (−117, 30)	6 (−236, 69)	−5 (−149, 37)
	via offspring’s overweight and height	19 (−12, 37)	5 (−16, 12)	27 (−18, 54)	6 (−21, 15)
at age 30 years but not before puberty	via offspring’s overweight	26 (−29, 57)	13 (−19, 29)	32 (−33, 67)	15 (−19, 34)
	via offspring’s height	61 (−104, 112)	39 (−87, 74)	91 (−151, 163)	50 (−110, 93)
	via offspring’s overweight and height	20 (−9, 42)	9 (−8, 21)	29 (−14, 60)	12 (−11, 27)

* Difference (beta) in offspring’s expected lung function value when the exposure is present, but offspring’s height and/or offspring’s overweight change from their expected value when the exposure is absent to their expected value when the exposure is present. Model 1 also includes the potential confounders and adjusting variables of the mediators (mothers’ low education level) and of the outcomes (mothers’ low education level and offspring’s age and ever smoking). 95% CI: 95% confidence interval.

## Data Availability

Requests for access to data can be made to the RHINESSA steering committee by PI Professor Cecilie Svanes (cecilie.svanes@uib.no) or vice PI Professor Vivi Schlünssen (vs@ph.au.dk). Reuse of the data must be done in collaboration with the RHINESSA study team. Further information including issues on data security and sharing of data can be found at www.rhinessa.net (accessed on 3 April 2022).

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
