# Peer review of "Parental Prepuberty Overweight and Offspring Lung Function"

_nutrients, 2022, doi:10.3390/nu14071506_

Round 1

Reviewer 1 Report

Svanes and co-workers propose a clinical study of causal effects of parents' overweight on their children's lung function. The study found that fathers' overweight before puberty appears to lower their future sons' FEV1 and FVC, but not mothers'. The effects were mediated partly by sons' adult height but not by sons' prepubertal obesity. These results are interesting, but this manuscript has some merit.

  1. Supplementary Materials aren't shown.
  2. In maternal lines, daughters had pre-bronchodilator FEV1 and FVC values that were statistically significantly lower (p-value <0.001) (Table 1). However, in Table 4, mothers’ overweight before puberty had direct effects on pre-bronchodilator FEV1 and FVC values.
  3. In the first presentation, the entire term must be presented. Examples include FEV and FVC.
  4. What is the protocol code for this study?

Reviewer 2 Report

I had the opportunity to review the manuscript entitled “Parental prepuberty overweight and offspring lung function”.

The paper is well written, with clearly method, results and discussion section. Authors used a sophisticated statistical methods, which allows to answer the questions stated in the aim of the study.
